# Constructing and Verifying an Alexithymia Risk-Prediction Model for Older Adults with Chronic Diseases Living in Nursing Homes: A Cross-Sectional Study in China

**DOI:** 10.3390/geriatrics7060139

**Published:** 2022-12-06

**Authors:** Jing Wen, Ying Wu, Lixia Peng, Siyi Chen, Jiayang Yuan, Weihong Wang, Li Cong

**Affiliations:** 1School of Medicine, Hunan Normal University, Changsha 410013, China; 2Shuda College, Hunan Normal University, Changsha 410000, China

**Keywords:** nursing home, older adults with chronic diseases, alexithymia, psychological resilience, risk prediction

## Abstract

Alexithymia is a critical global public health concern. This questionnaire-based cross-sectional study explored the risk factors of alexithymia in older adults living in nursing homes with chronic diseases. It also developed and evaluated an alexithymia risk-prediction model. A total of 203 older adults with chronic diseases were selected from seven nursing homes in Changsha, China, using simple random and cluster sampling. The participants were surveyed using the Toronto Alexithymia Scale (TAS-20), Geriatric Depression Scale-15 (GDS-15), Connor-Davidson Resilience Scale (CD-RISC), Perceived Social Support Scale (PSSS), and a socio-demographic characteristics questionnaire. The alexithymia total score was 43.85 ± 9.570, with an incidence rate of 8.9%. Alexithymia had a partial mediating effect on the relationship between social support and psychological resilience (the effect value was 0.12), accounting for 19.04% of the total effect. Gender, depression, and psychological resilience were the main independent influencing factors of alexithymia (*p* < 0.05). The area under the receiver operating characteristic (AUC-ROC) curve of the risk-prediction model was 0.770. The participants, especially those who were male and depressed, exhibited a certain degree of alexithymia. Additionally, it partially mediated the association between social support and psychological resilience, which is a protective factor against alexithymia. The risk-prediction model showed good accuracy and discrimination. Hence, it can be used for preliminary screening of alexithymia in older adults with chronic diseases living in nursing homes.

## 1. Introduction

According to the latest information released by the National Bureau of Statistics of China in 2020, 18.7% of the country’s total population is over 60 years old and 13.5% is over 65 years old [1]. Owing to traditional Chinese culture, family care is still the primary mode of care, but the rapidly aging population has intensified the burdens on the family. Further, the demand for aged care is constantly on the rise, making nursing homes an indispensable part of the Chinese multi-level elder care service system. Deteriorating physiological functions among older adults largely increase the risk of chronic diseases, which not only cause physical pain, but also seriously affect psychological health due to their persistence and complexity [2]. Moreover, changes in the living environment, lifestyle, and social relationships may make older adults living in institutions resentful toward communicating with others. They may eventually become depressed and reluctant to express their emotions, potentially leading to encounter difficulties and psychological deterioration.

Alexithymia is derived from Greek, and means “no words for emotions” [3]. It is a psychological problem characterized by affective cognitive disorder, which is marked by the inability to describe one’s own emotions or recognize others’ feelings [4]. It can be classified into primary or secondary alexithymia, depending on the cause. Primary alexithymia is generally not considered to be a separate mental illness but a personal trait; secondary alexithymia entails psychological distress from traumatic events [5]. Patients with alexithymia tend to confuse somatosensory and emotional experiences, affecting their treatment and rehabilitation, aggravating their psychological disorders, and severely damaging their physical and mental health, especially in cases of aging patients with chronic diseases [6]. Hence, studying its influencing factors is practically significant because it could help in the early screening of alexithymia and targeted interventions for the entire life-cycle management of older adults with chronic diseases.

Resilience, which represents an individual’s ability to positively adapt to and recover from adversity or hardship, is defined as a protective and salutogenic factor [7]. Perceived social support is the quantity and quality of material and spiritual support subjectively perceived by individuals from their social networks [8]. One study showed that perceived social support positively affects the protection and maintenance of psychological health and can directly affect individual psychological resilience [9]. Furthermore, perceived social support and psychological resilience are negatively correlated with alexithymia, and effective social support significantly reduces the incidence of alexithymia and depression [10]. These findings indicated that alexithymia, psychological resilience, and perceived social support are related to each other. However, research aimed at exploring the potential mechanisms underlying their association remains limited.

Alexithymia is crucial in terms of older adults’ physical and mental health, especially for those with chronic diseases who live in nursing homes. Despite this, currently, the research on alexithymia in older adults with chronic diseases in Chinese nursing homes is restricted to a status survey and influencing factors analysis. There are no reliable risk-prediction models. Therefore, this study explored the relationship between alexithymia, psychological resilience, and perceived social support using a construction equation model. Additionally, it constructed an alexithymia risk-prediction model based on the major independent influencing factors. Knowledge about the risk factors of alexithymia and moderating mechanisms underlying the association between alexithymia, psychological resilience, and perceived social support could help in the early detection and prevention of alexithymia. It could also elucidate the implications of improving mental health that could further lead to the development of more effective strategies for older adults with chronic diseases who live in nursing homes.

## 2. Methods

### 2.1. Study Design

This study implemented a cross-sectional design to explore the mechanisms of alexithymia, resilience, and perceived social support through structural equation modeling, and to construct a risk-prediction model to understand the risk factors associated with alexithymia in older adults with chronic diseases in nursing homes. This study was undertaken in adherence to the STROBE Statement for observational research [11].

### 2.2. Participants

We first numbered the nursing homes in the four districts of Changsha separately. Subsequently, we randomly selected one to two nursing homes from each district, and seven nursing homes were included. Participants who met the inclusion criteria were recruited in March 2021 using the cluster sampling method. Those with at least one chronic disease, clear consciousness, reading or verbal ability, and barrier-free communication who provided informed consent and voluntarily participated were included. Those with severe physical or mental illness were excluded. When using multifactor analysis, the sample size should be 5 to 10 times the number of variables or the number of scale dimensions [12]. Twenty-three variables were studied, and hence, the required sample size was 115–230 in this study. A total of 221 older adults with chronic diseases from the seven nursing homes met the inclusion and exclusion criteria. We invited these 221 older adults to participate in the survey separately, but after describing the purpose and methods of this study, 18 of them refused to participate, and 203 were finally included in this study. 

### 2.3. Socio-Demographic Characteristics Questionnaire

The socio-demographic characteristics questionnaire was designed by the investigators based on relevant publications and research. It contains 16 entries: age, gender, number of children, time spent in the nursing home, marital status, occupation, education level, location of the nursing home, main economic sources, main reason for staying in the nursing home, frequency of friends’ or relatives’ visits, whether participants went back home during festivals, self-care ability, satisfaction with food, accommodation, and caregivers in nursing homes.

### 2.4. Toronto Alexithymia Scale (TAS-20)

TAS-20 was developed by Bagby [13] in 1994, and the Chinese version was revised by Yi [14] to make it suitable for Chinese respondents. The scale comprises three subscales: difficulty identifying feelings (DIF), difficulty describing feelings (DDF), and externally oriented thinking (EOT). A 5-point Likert-type scale was used, reverse-scoring for 5 entries (entries 4, 5, 10, 18, 19) and forward-scoring for others. The total score ranges from 20 to 100, where higher scores indicate a higher extent of alexithymia. The Cronbach’s α coefficient and retest reliability of this scale are 0.83 and 0.87, respectively.

### 2.5. Geriatric Depression Scale-15 (GDS-15)

GDS-15 was originally developed by Brink et al. [15], which contained 30 items. Sheikh et al. [16] adjusted and revised the scale to contain only 15 items and make it more suitable for older adults. The response of “1” indicates “yes” and “0” indicates “no”. A cutoff value between 0–4 is considered as “no depression”, 5–10 as “mild depression”, and 11–15 as “severe depression”. The higher the score, the more serious the depressive symptoms. The Cronbach’s α coefficient of this scale is 0.846, and the retest reliability is 0.812.

### 2.6. The Connor-Davidson Resilience Scale (CD-RISC)

CD-RISC was compiled by Connor and Davidson [17], and the Chinese version was revised by Yu [18]. This scale has 25 items with 3 factors, namely tenacity, strength, and optimism, and it is rated on a 5-point Likert scale (0 = not true at all to 4 = true all the time). The total score is 0–100, and the higher the score, the greater the psychological resilience. The Cronbach’s α coefficient of CD-RISC and its subscales are 0.779, 0.819, 0.803, and 0.652, respectively.

### 2.7. Perceived Social Support Scale (PSSS)

PSSS was compiled by Zimet et al. [19] in 1988 and revised into a Chinese version by Jiang et al. [20] It is used to assess the degree of social support from three perspectives: family support, friend support, and other support. The questionnaire includes 12 items and is rated on a 7-point Likert scale (1 = strongly disagree to 7 = strongly agree). The total score is 12 to 84, with a higher score indicating a higher level of perceived social support. The Cronbach’s α coefficient of the total amount table and three dimensions are 0.836, 0.705, 0.599, and 0.820, respectively.

### 2.8. Data Collection

This study adopted the questionnaire survey method. All participants provided their signed informed consent. Prior to consenting, they were informed about the study in a written format. The questionnaires were sent over to the seven selected nursing homes for data collection and were filled in by the participants in separate rooms. Those with dysgraphia were assisted by trained investigators. After completing the investigation, all questionnaires were submitted on site and subsequently examined individually. In cases of missing or repeated information, the participants were asked to supplement and modify the answers. All 203 of the questionnaires that were sent out were received, resulting in a 100% response rate.

### 2.9. Statistical Analysis

All statistical analyses were performed using SPSS 22.0 (IBM Corp; Armonk, NY, USA) and AMOS 21.0 (IBM Corp; Armonk, NY, USA). The *t*-test and one-way analysis of variance (ANOVA) compared the demography-based differences in alexithymia. Pearson correlation was used to examine the association between TAS-20, GDS-15, CD-RISC, and PSSS. Structural equation modeling and path analysis considered PSSS as the dependent variable, CD-RISC as the independent variable, and TAS-20 as the intermediary variable. The bootstrap method tested the significance of mediation effects, and the maximum likelihood method was used to modify and fit the model. When the bootstrap 95% confidence interval did not contain 0, the mediating effect was considered to be statistically significant. Mediating effects are classified as fully mediated and partially mediated. Full mediation implies that the mediating variable M explains the whole relationship between the independent variable X and dependent variables Y, while partial mediation implies that the mediating variable M explains some, but not all, of the relationship between the independent and dependent variables. Subsequently, multiple regression analysis was used to detect the risk factors of alexithymia. R software (Vienna, Austria version 3.6.3) was used for developing the risk-prediction model, which was visualized through a nomogram. The predictive efficiency was evaluated based on the AUC-ROC curve. All statistical tests were two-tailed, and the statistical significance for all analyses was set at 0.05.

## 3. Results

### 3.1. Participants’ Characteristics and Univariate Analysis of Alexithymia 

Among the 203 participants, 81 were male and 122 were female. The average age was 81.59 ± 8.97 years. The alexithymia total score was 43.85 ± 9.570; 174 (85.7%) had no alexithymia, 11 (5.4%) exhibited a tendency, and 18 (8.9%) showed obvious signs of alexithymia. The scores for each aspect of alexithymia, from low to high, were 11.15 ± 5.439 for difficulty identifying feelings, 12.93 ± 2.797 for difficulty describing feelings, and 19.77 ± 4.236 for externally oriented thinking. The results of univariate analysis (Table 1) indicated that those who were female, who lived in rural areas, and who could take care of themselves had a lower degree of alexithymia (*p* < 0.05). In other words, there were demography-based differences in the degree of alexithymia. Thus, special groups should be given more attention and personalized support strategies.

### 3.2. Correlation Analysis of Alexithymia, Depression, Psychological Resilience, and Social Support

Pearson correlation analysis showed that alexithymia was positively correlated with GDS-15 (r = 0.300, *p* < 0.01) and negatively correlated with CD-RISC and PSSS (r = −0.315, −0.182, *p* < 0.01), especially difficulty identifying feelings and externally oriented thinking (Table 2). Accordingly, we speculated that depression may be a risk factor for alexithymia, and psychological resilience and social support may be protective factors that mainly affect emotional recognition and external thinking.

### 3.3. Mediating Effect Analysis of Alexithymia, Psychological Resilience, and Perceived Social Support

A structural equation model was used to explore the mediating effect of alexithymia on the relationship between psychological resilience and perceived social support. The model-fitting index showed that alexithymia played a partially mediating role, with a mediating effect value of 0.12, accounting for the total effect of 19.04% (0.120/0.630 × 100% = 19.04%). Additionally, the fitting indexed χ^2^ = 56.523, df = 28, χ^2^/df = 1.764 < 3, RMSEA = 0.061 < 0.08, CFI = 0.964 > 0.90, TLI = 0.967 > 0.80, NFI = 0.960 > 0.80, and GFI = 0.964 > 0.08, which indicated that the intermediary model (Figure 1) was reasonable. The bootstrap method with bias correction was used to test the significance of the mediating effect; the 95% confidence interval (CI) of the total effect was 0.42, 0.83; of the direct effect was 0.27, 0.71; and of the indirect effect was 0.03, 0.29. The mediating effect was statistically significant (*p* < 0.01). Results of mediation analyses revealed that alexithymia played a mediating role in the relationship between psychological resilience and perceived social support.

### 3.4. Analysis of the Influencing Factors of Alexithymia

The alexithymia total score was taken as the dependent variable and GDS-15, CD-RISC, PSSS, gender, location of nursing home, and self-care ability as independent variables in the multiple regression analysis. The results showed that gender, depression, and psychological resilience enter the regression model (Table 3). Further, the female gender and psychological resilience were protective factors, and depression was a risk factor.

### 3.5. Construction of an Alexithymia Risk-Prediction Model

Based on the independent influencing factors (gender, Geriatric Depression, and Connor-Davidson resilience) entering the regression model, the alexithymia risk-prediction model was $\hat{Y}$ = 59.44 − 03.297 × gender + 0.493 × GDS-15 − 0.138 × CD-RISC. This study constructed a nomogram model for older patients with chronic diseases and alexithymia in care institutions based on the factors mentioned above (Figure 2). According to the nomogram, the alexithymia incidence of patients can be quickly predicted. The ROC curve analysis showed that the AUC of the nomogram model predicting senile chronic disease patients with alexithymia is 0.770 (95% CI: 0.691–0.850). AUC below 0.6 is considered low discrimination, 0.6–0.75 is medium discrimination, and above 0.75 is high discrimination. Thus, this nomogram model had good discrimination (Figure 3).

## 4. Discussion

This study investigated the relationship between alexithymia, individual psychological resilience, and perceived social support in older adults with chronic diseases residing in Chinese nursing homes. The structural equation model indicated that higher perceived social support was associated with higher psychological resilience, and this relationship was partially mediated by alexithymia. The risk-prediction model revealed gender (male) and depression as risk factors for alexithymia; psychological resilience was a protective factor.

The incidence of alexithymia in nursing homes in this study was 8.9%, which was lower than that among the community-dwelling and empty-nest older adults in China [21]. This may be related to the professional care and convenient medical and health services provided by nursing homes. In descending order, the alexithymia-related scores pertained to externally oriented thinking, difficulty describing feelings, and difficulty identifying feelings. The results indicated that the study participants could describe their emotions in words, but they did not pay attention to their feelings and imagination-related activities and paid more attention to external details. They also had difficulty identifying their own and others’ emotions and distinguishing between physical symptoms and emotional problems.

The score and incidence of alexithymia in men were higher than in women, which is consistent with Jan and John, respectively [22,23]. This may be related to education or personality. Men generally were introverted and unwilling to share or vent their negative emotions. Suppressing such negative emotions over a long period of time made it difficult for them to identify and describe their feelings, finally culminating in alexithymia. Clearly, nursing-home workers should pay more attention to the mental health of older men, encourage them to actively communicate, and express their inner feelings. It is also important to identify, express, and release emotions promptly, strengthen psychological care and emotional support, and conduct positive interventions to reduce the incidence of alexithymia.

Alexithymia was a susceptibility factor for the occurrence and/or continuance of several mental diseases, such as anxiety and depression. High-level alexithymia was more likely to lead to depression [24,25]. This especially held true in cases where long-term negative experiences and emotions owing to life’s difficulties could not be fully vented; such a situation aggravated depression [26]. Moreover, those with depression could not correctly use functional methods to recognize emotions, so they had more difficulties in subjectively identifying and describing their emotions. We found that depression was positively correlated with alexithymia and was a risk factor. Therefore, when depressed older patients turn to medication, the managers of nursing homes should conduct psychotherapy to aid such patients in effectively regulating, controlling, and releasing emotions. These measures would also provide an opportunity for older adults to focus on their physical and psychological conditions, acquire knowledge about mental health, promote emotional exchanges, and decrease alexithymia.

Furthermore, resilience is a dynamic structure that involves adaptability, sustainability, and recovery in the face of stress or trauma throughout one’s life cycle. Studies have shown that a decline in psychological resilience may cause emotional anxiety and alexithymia in patients with functional neurological disorders [27,28]. A higher level of psychological resilience can help individuals recover from adversity and maintain or improve mental health [29]. Our results showed that all aspects of psychological resilience were significantly negatively correlated with alexithymia. Therefore, psychological flexibility training should be conducted in nursing homes, such as holding regular educational seminars, evaluating psychological status by professional psychological counselors, and conducting psychological counseling. This can help older patients in nursing homes build strong psychological resilience, promptly adapt to the environment, better manage diseases, and improve their quality of life.

In our study, the mediating effect of alexithymia in the relationship between social support and psychological resilience was discussed through the structural equation model. Alexithymia negatively predicted psychological resilience, but social support positively predicted psychological resilience. Social support directly affected mental health by enhancing the sense of goals, belonging, and security [30]. Individuals with alexithymia have defects in their ability to express themselves and cannot accurately describe their emotions. When they encounter difficulties in interpersonal interactions or life, they are unable to confide in and share their emotions with others, resulting in increasing negative energy, which affects resilience when it reaches a certain level [31]. Therefore, the managers and caregivers in nursing homes should encourage older patients’ families to contact them, enrich daily activities that can help them make new friends, let them feel loved, reduce loneliness, and improve such patients’ social support system.

The essence of the nomogram is the visualization of a regression equation, which displays the regression results through a flat graph with scale line segments [32]. This study integrated three variables that were entered into the regression equation model to construct a nomogram model. In our study, the AUC of the alexithymia risk-prediction model was 0.770, indicating that the nomogram entailed good discrimination. Therefore, the nomogram based on the risk-prediction model could provide a relatively personalized and accurate risk estimate of the probability of alexithymia among older adults with chronic diseases in nursing homes.

## 5. Limitations

This study has several limitations. Due to minimal resources such as personnel and funds, this study only focused on older adults with chronic diseases in nursing homes in Changsha. Second, the sample was relatively small, which may affect the reliability and comprehensiveness of this study; additionally, the prediction model was not externally validated. Third, we used a cross-sectional study design, which, compared to longitudinal studies, could not accurately determine the causal relationship. Finally, this study did not specifically classify chronic non-infectious diseases in the sample selection process, which may result in the absence of specificity among the population using the prediction model. In the future, we can increase the sample size and select nursing homes in different regions to enrich and improve the results. A longitudinal follow-up survey could be launched to obtain data and verify the causal relationship between variables. Subsequently, alexithymia can be analyzed more rationally and objectively, which would further promote the physical and mental well-being of older adults with chronic diseases in nursing homes.

## 6. Conclusions

Based on the TAS-20, GDS-15, PSSS, and CD-RISC, this study analyzed the current situation and influencing factors of alexithymia among older adults with chronic diseases living in nursing homes. The incidence of alexithymia was relatively high and had a mediating effect on the relationship between social support and psychological resilience. Gender, depression, and psychological resilience were independent influencing factors of alexithymia. The risk-prediction model we developed showed good predictive efficacy and could be used to assess the risk of alexithymia in older adults. In this study, by building a model, the protective factors and risk factors related to alexithymia in older adults with chronic diseases in nursing homes were understood to a certain extent, so as to reduce the occurrence of alexithymia and improve their psychological resilience.

## Figures and Tables

**Figure 1 geriatrics-07-00139-f001:**
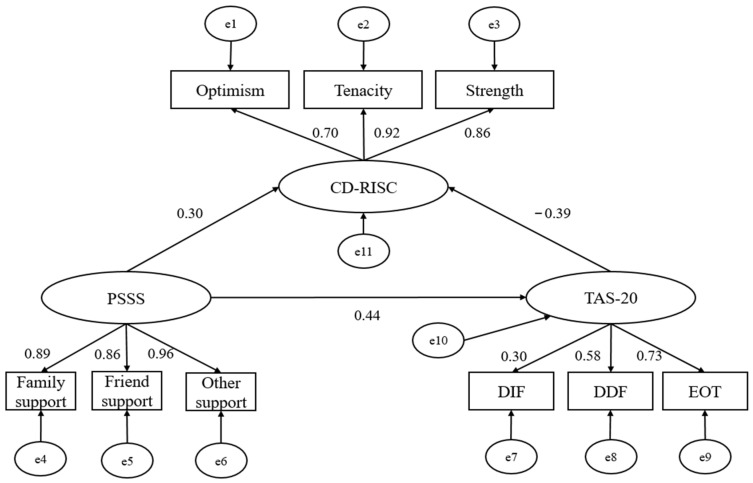
The mediating path of alexithymia between psychological resilience and perceived social support. PSSS: Perceived Social Support Scale; TAS-20: Toronto Alexithymia Scale; CD-RISC: Connor-Davidson Resilience Scale; DIF: difficulty identifying feelings; DDF: difficulty describing feeling; EOT: externally oriented thinking; e1–11: error 1–11, arrow: pathway, number: standardized estimates.

**Figure 2 geriatrics-07-00139-f002:**
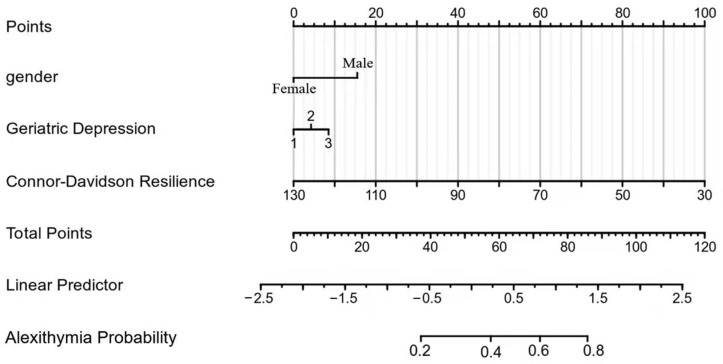
Nomogram of the alexithymia risk-prediction model.

**Figure 3 geriatrics-07-00139-f003:**
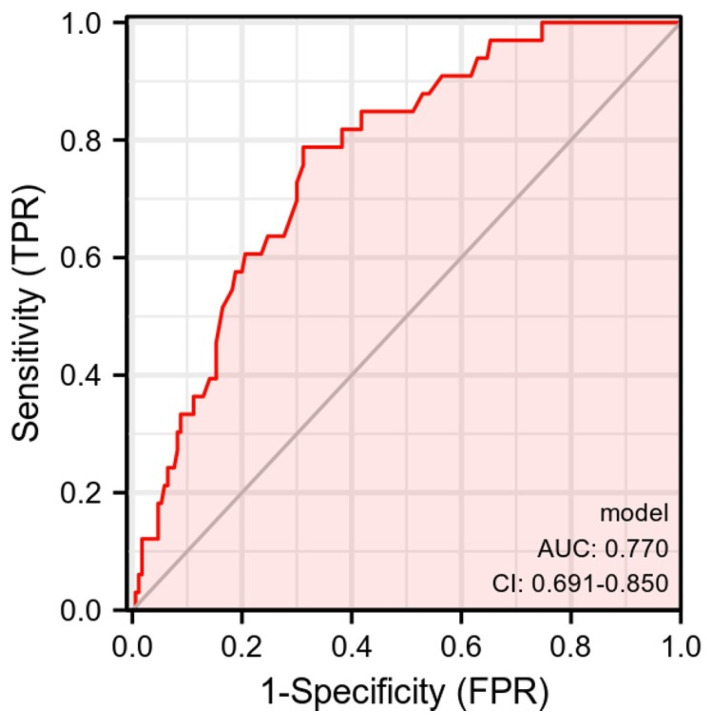
ROC curve of the alexithymia risk-prediction model.

**Table 1 geriatrics-07-00139-t001:** General information of study participants and univariate analysis of alexithymia (n = 203).

Variables	n %	M ± SD	*F*/*t*	*p*-Value
Age (years)			0.979	0.42
≤60	4 (2.0)	46.00 ± 13.88		
61~70	28 (13.8)	45.39 ± 7.42		
71~80	49 (24.1)	45.20 ± 10.53		
81~90	94 (46.3)	42.47 ± 9.75		
≥91	28 (13.8)	44.25 ± 8.41		
Gender			2.745	0.007
Male	81 (39.9)	46.07 ± 10.14		
Female	122 (60.1)	42.37 ± 8.91		
Marital status			1.111	0.346
Single	3 (1.5)	43.33 ± 4.93		
Married	51 (25.1)	44.41 ± 10.94		
Widowed	139 (68.5)	43.29 ± 9.10		
Separated	10 (4.9)	48.8 ± 9.10		
Number of children			0.563	0.64
0	10 (4.9)	44.30 ± 4.72		
1	38 (18.7)	43.16 ± 8.28		
2	87 (42.9)	44.80 ± 10.48		
≥3	68 (33.5)	42.94 ± 9.60		
Time in nursing homes (years)			2.089	0.127
<1	53 (26.1)	42.04 ± 8.70		
1~5	91 (44.8)	45.27 ± 10.95		
≥5	59 (29.1)	43.27 ± 7.67		
Occupation			2.11	0.081
Farmer	9 (4.4)	47.89 ± 9.01		
Self-employed households	13 (6.4)	46.62 ± 10.74		
Staff	50 (24.6)	45.80 ± 9.34		
Institutional personnel	96 (47.3)	43.07 ± 9.70		
Others	35 (17.2)	41.11 ± 8.56		
Education level			2.617	0.052
Primary school	43 (21.2)	46.12 ± 8.96		
Junior high school	61 (30.0)	42.89 ± 8.60		
High school	44 (21.7)	45.73 ± 10.65		
Bachelor/Master/PhD	55 (27.1)	41.64 ± 9.73		
Location of nursing home			−2.03	0.044
Countryside	32 (15.8)	40.72 ± 9.05		
City	171 (84.2)	44.43 ± 9.58		
Main economic sources			0.879	0.453
Children	59 (29.1)	45.27 ± 9.51		
Government	2 (1.0)	41.50 ± 0.71		
Own	138 (68.0)	43.41 ± 9.74		
Others	4 (2.0)	39.25 ± 2.50		
Main reason for staying in the nursing home			0.705	0.495
No children	9 (4.4)	46.44 ± 8.17		
Reduce burden on children	57 (28.1)	44.63 ± 9.27		
Receive professional care	137 (67.5)	43.35 ± 9.79		
Frequency of friends’ or relatives’ visits			1.395	0.228
Never	8 (3.9)	40.38 ± 11.26		
Weekly	91 (44.8)	45.73 ± 10.49		
Fortnightly	31 (15.3)	42.94 ± 8.70		
Triweekly	4 (2.0)	41.25 ± 4.79		
Monthly	23 (11.3)	42.70 ± 7.35		
More than a month	46 (22.7)	42.15 ± 8.89		
Back home during festivals			1.069	0.286
Yes	97 (47.8)	44.60 ± 9.72		
No	106 (52.2)	43.16 ± 9.43		
Self-care ability			5.22	0.006
Yes	116 (57.1)	42.12 ± 8.42		
Partly	81 (39.9)	46.46 ± 10.70		
No	6 (3.0)	42.00 ± 7.32		
Satisfaction with accommodation			0.967	0.427
Very dissatisfied	1 (0.5)	40.50 ± 11.62		
Dissatisfied	4 (2.0)	38.25 ± 6.95		
Neutral	25 (12.3)	45.08 ± 8.03		
Satisfied	117 (57.6)	44.53 ± 10.74		
Very satisfied	56 (27.6)	42.55 ± 7.41		
Satisfaction with food			1.346	0.254
Very dissatisfied	9 (4.4)	43.44 ± 14.43		
Dissatisfied	34 (16.7)	43.47 ± 9.30		
Neutral	42 (20.7)	41.50 ± 6.57		
Satisfied	94 (46.3)	45.37 ± 10.74		
Very satisfied	24 (11.8)	42.67 ± 6.55		
Satisfaction with caregivers			0.776	0.542
Very dissatisfied	5 (2.5)	41.00 ± 8.52		
Dissatisfied	8 (3.9)	47.00 ± 9.30		
Neutral	22 (10.8)	41.95 ± 8.64		
Satisfied	102 (50.2)	44.59 ± 10.54		
Very satisfied	66 (32.5)	43.17 ± 8.35		

Note: M: mean, SD: standard deviation.

**Table 2 geriatrics-07-00139-t002:** Correlation analysis of alexithymia, depression, psychological resilience, and social support.

Variables	DIF	DDF	EOT	Alexithymia
GDS-15	0.364 **	0.129	0.122	0.300 **
CD-RISC	−0.353 **	0.016	−0.264 **	−0.315 **
Tenacity	−0.375 **	0.015	−0.261 **	−0.327 **
Strength	−0.319 **	0.038	−0.232 **	−0.275 **
Optimism	−0.156 *	−0.029	−0.192 **	−0.184 **
PSSS	−0.164 *	0.083	−0.250 **	−0.182 **
Family support	−0.080	0.139 *	−0.152 *	−0.074
Friend support	−0.277 **	0.016	−0.311 **	−0.293 **
Other support	−0.053	0.063	−0.181 **	−0.093

Note: DIF: difficulty identifying feelings; DDF: difficulty describing feelings; EOT: externally oriented thinking. * *p* < 0.05, ** *p* < 0.01.

**Table 3 geriatrics-07-00139-t003:** Regression analysis of influencing factors of alexithymia.

	B	SE	Β	*t*	*p*-Value
Constant	59.440	4.631		11.504	<0.001
Gender	−3.297	1.270	0.133	−2.596	0.010
GDS-15	0.493	0.189	0.383	2.601	0.010
CD-RISC	−0.138	0.043	0.335	−3.234	0.001

Note: R = 0.402, R^2^ = 0.161, R^2^_adj_ = 0.149; SE: standard error.

## Data Availability

The data presented in this study are available on request from the corresponding author. The data are not publicly available due to privacy.

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
