# Peer review of "Constructing and Verifying an Alexithymia Risk-Prediction Model for Older Adults with Chronic Diseases Living in Nursing Homes: A Cross-Sectional Study in China"

_geriatrics, 2022, doi:10.3390/geriatrics7060139_

Round 1

Reviewer 1 Report

Thank you for this very interesting manuscript. While alexithymia may not be viewed as a serious problem in an aging population, the authors demonstrate its impact on a segment of the elderly people in nursing homes. It is interesting that the rate of alexithymia is lower in this population. The key contribution of this work is the mapping of the role alexithymia plays in relation to resilience and social support. The risk assessment is novel and will need verification.

Reviewer 2 Report

This manuscript should be publishable with some revisions.

It is important to verify that assumptions underlying linear models are satisfied regarding normality, homoscedasticity, and linearity. Randomization has been verified.

Some further explanation regarding the ROC would be helpful. It is presented with very little explanation.

When editing the manuscript to enhance clarity, please be careful about the use of terms such as "elderly," "elderlies," and similar terms that are derogatory.

Reviewer 3 Report

I would like to congratulate the authors for their study on an interesting and relevant subject. Ageing is a period characterized by physical, affective and economics loses, where alexithymia seems to be a common feature of neurological disease. Thus, detecting alexithymia could be relevant for preserving the wellbeing of older adults

 In the present manuscript authors intended to explore the relationship between alexithymia, perceived social support and psychological resilience in older adults with chronic diseases living in nursing homes. Moreover, they intend to construct an alexithymia risk prediction model based on major independent influencing factors, to provide references for early identification and prevention of alexithymia

The research presented in this manuscript is both technically and ethically sound, and the data supports the main conclusion.

Considering the above, I only have some minor revisions:

1)      The replacement of the expression elderly people and senior by “older adults”

2)      Line 125-127 is not clear by the analysis of the table. A more detailed description should be helpful

Figure 2 a more complete description of the nomogram of alexithymia is mandatory in order to promote a better understanding

Reviewer 4 Report

The authors conducted a cross-sectional observational study to examine risk factors of alexithymia in elderly patients with chronic diseases in nursing homes and to develop the risk prediction model. The authors analyzed data from 203 elderly patients with chronic diseases from 7 nursing homes in Changsha. They showed that alexithymia mediated the effects of social support on psychological resilience. They then constructed a prediction model with alexithymia as the dependent variable and psychological resilience, gender, and depression (GDS-15) as the independent variables. They showed that the AUC of the model was 0.77.

There are some comments.

Major Comments:

1.      The authors aimed to examine alexithymia's risk factors and to develop a model predicting alexithymia. The aims require a longitudinal study. However, a cross-sectional study was conducted.

2.      The structural equation model showed that alexithymia mediated the effects of social support on psychological resilience. In other words, alexithymia influenced psychological resilience rather than vice versa. These results support constructing a model that predicts psychological resilience using alexithymia. However, the results do not support constructing a model that predicts alexithymia using psychological resilience.

3.      This study examined alexithymia in older patients with chronic diseases. However, diseases were not assessed.

Minor Comments:

4.      Introduction: This study examined multiple factors, including socio-demographic factors and depressive mood, as potential influencing (risk) factors of alexithymia. However, a rationale for examining these factors is lacking.

5.      Introduction (Line 33): "lead to the emergence and deterioration of psychological problems." Why, then, would the authors like to develop a model predicting alexithymia using psychological problems (depressive mood)?

6.      Methods (Participants): The study included 203 elderly patients with chronic diseases from 7 nursing homes in Changsha. It is unclear how many subjects were screened for eligibility, how many screened subjects were eligible, and how many eligible subjects agreed to enter the study.

7.      Methods (Statistical analysis, Line 111): The authors conducted a structural equation model with PSSS as the dependent variable, CD-RISC as the independent variable, and TAS-20 as the intermediary variable. However, the rationale was not explained (for instance, in the Introduction). In addition, it is unclear how the results of such an analysis would justify the use of PSSS and CD-RISC in predicting alexithymia.

8.      Results (Table 1): Was "X±s" the mean (sd) TAS-20 score? An explanation is recommended.

9.      Results (Figure 1): A more detailed explanation of the graph in the figure legends is recommended. For instance, what did the arrow mean? What did the number mean? What did "e" stand for?

10.  Discussion: A summary of the key findings of this study at the beginning of the first paragraph is recommended.

11.  Discussion (Line 226-227): "Patients who have trouble describing emotions may encounter difficulties in interpersonal communication, which in turn lead to decline in social ability and reduction in seeking social support." However, this contradicted the results of the structural equation model.

12.  Discussion: A major limitation of this study is that it is cross-sectional, which precluded the estimation of causal effects (including mediating effects), and the temporal relationships between alexithymia, perceived social support, and psychological resilience. A discussion is recommended.

13.  Abstract: Conclusion is lacking.

Round 2

Reviewer 4 Report

The authors have addressed the issues.

Author Response

Thank you again for your time, effort, and constructive comments, which helped us improve and perfect our manuscript.